# TransAdapter: Vision Transformer for Feature-Centric Unsupervised Domain Adaptation

## Abstract

Unsupervised Domain Adaptation (UDA) aims to leverage labeled data from a source domain to address tasks in a related but unlabeled target domain. This problem is particularly challenging when there is a significant gap between the source and target domains. Traditional methods have largely focused on minimizing this domain gap by learning domain-invariant feature representations using convolutional neural networks (CNNs). However, recent advances in vision transformers, such as the Swin Transformer, have demonstrated superior performance in various vision tasks. In this work, we propose a novel UDA approach based on the Swin Transformer, introducing three key modules to improve domain adaptation. First, we develop a Graph Domain Discriminator that plays a crucial role in domain alignment by capturing pixel-wise correlations through a graph convolutional layer, operating on both shallow and deep features in the transformer. This module also calculates the entropy for the key attention features of the attention block to better distinguish between the source and target domains. Second, we present an Adaptive Double Attention module that simultaneously processes Windows and Shifted Windows attention to increase long-range dependency features. An attention reweighting mechanism is employed to dynamically adjust the contributions of the attention values, thereby improving feature alignment between domains. Finally, we introduce Cross-Feature Transform, where random Swin Transformer blocks are selectively transformed using our proposed transform module, enhancing the model's ability to generalize across domains by transferring the source to the target features. Extensive experiments demonstrate that our method improves the state-of-the-art on several challenging UDA benchmarks, confirming the effectiveness of our approach. In particular, our model does not include a task-specific domain alignment module, making it more versatile for various applications.

## 1 Introduction

Deep neural networks (DNNs) have achieved remarkable success in various machine learning tasks, particularly in computer vision (Wang et al., 2022a; Qian et al., 2021; Jiang et al., 2022; Tan et al., 2019; Chen et al., 2021b; Jiang et al., 2021). However, their performance often relies on large amounts of labeled training data, which can be costly and time-consuming to collect (Csurka, 2017; Zhao et al., 2020; Zhang et al., 2020; Oza et al., 2021). To address this, Unsupervised Domain Adaptation (UDA) has emerged as a viable alternative, transferring knowledge from a labeled source domain to an unlabeled target domain and mitigating challenges posed by domain shifts (Bousmalis et al., 2017; Kuroki et al., 2019; Wilson & Cook, 2020; VS et al., 2021).

Traditional UDA methods have used Convolutional Neural Networks (CNNs) to align source and target domains by learning transferable features across varying distributions (Kang et al., 2019; Zhang et al., 2019; Jiang et al., 2020; Li et al., 2021b). While these methods have made strides in reducing domain discrepancies through adversarial training and feature normalization, they can struggle with complex domain shifts and variability in visual patterns, highlighting ongoing

challenges in domain generalization and cross-domain alignment (Morerio et al., 2020; Jiang et al., 2020).

The advent of transformers has revolutionized feature learning in both natural language processing (NLP) (Vaswani et al., 2017; Devlin et al., 2018) and computer vision (Dosovitskiy et al., 2020; Han et al., 2020; He et al., 2021; Khan et al., 2021). The Swin Transformer (Liu et al., 2021) excels in modeling long-range dependencies and processes images in non-overlapping patches, enabling effective localized adaptation. This multiscale approach is well-suited for UDA tasks, ensuring robust feature representation and precise domain alignment.

This work introduces TransAdapter, a novel framework leveraging the Swin Transformer for UDA. It addresses the limitations of traditional CNN-based methods by incorporating three key modules: a Graph Domain Discriminator, an Adaptive Double Attention module, and a Cross-Feature Transform module. These components enhance domain adaptation performance by facilitating better alignment and improving local and global feature consistency across domains.

Contributions of this paper are summarized as follows:

- The Graph Domain Discriminator captures both shallow and deep features using graph convolutional layers, enhancing pixel-wise correlation and domain alignment. we incorporate entropy in the key attention features, preventing the attention mechanism from focusing too narrowly on specific regions, leading to more balanced and transferable representations.

- The Adaptive Double Attention module captures long-range dependencies by simultaneously processing window and shifted window attention. This dual mechanism maintains global and local features, while an attention reweighting module enhances feature alignment and overall model performance.

- The Cross Feature Transform module adapts the Swin Transformer for UDA tasks. By randomly selecting a transformer block and applying a specialized transform module in each iteration, the model dynamically explores different aspects of the feature space. This enhances domain adaptation and improves performance across diverse datasets.

In summary, integrating these three modules within the Swin Transformer framework provides a robust solution for UDA, effectively addressing domain shift challenges and advancing the state-of-the-art in domain adaptation for computer vision tasks.

## 2 RELATED WORK

### 2.1 UNSUPERVISED DOMAIN ADAPTATION (UDA) AND TRANSFER LEARNING

Unsupervised Domain Adaptation (UDA) within transfer learning aims to learn transferable knowledge that is generalizable across different domains with varying data distributions. The main challenge lies in addressing the domain shift the discrepancy in the probability distributions between source and target domains.

Early UDA methods, such as Deep Domain Confusion (DDC), focused on learning domain-invariant characteristics by minimizing the maximum mean discrepancy (MMD) between two domains (Tzeng et al., 2014a). This helped align the marginal distributions of the source and target domain data at a feature level. Long et al. (Long et al., 2015b) enhanced this approach by embedding hidden representations in a reproducing kernel Hilbert space (RKHS) and applying a multiple-kernel variant of MMD to measure the domain distance more effectively. These hidden representations refer to the activations within layers of a deep neural network, where each layer captures different hierarchical features of the input data.

To further improve alignment, Long et al. (Long et al., 2017) proposed aligning the joint distributions of multiple domain-specific layers across domains using a joint maximum mean discrepancy (JMMD) metric. These layers refer to the different layers in a neural network, where each layer encodes various aspects of the data, from low-level features in earlier layers to high-level semantic information in deeper layers. The idea is to align not only the marginal distributions of individual layers but also the joint distribution of features across multiple layers, ensuring that both lower-level and higher-level representations are aligned between domains. Adversarial learning methods,

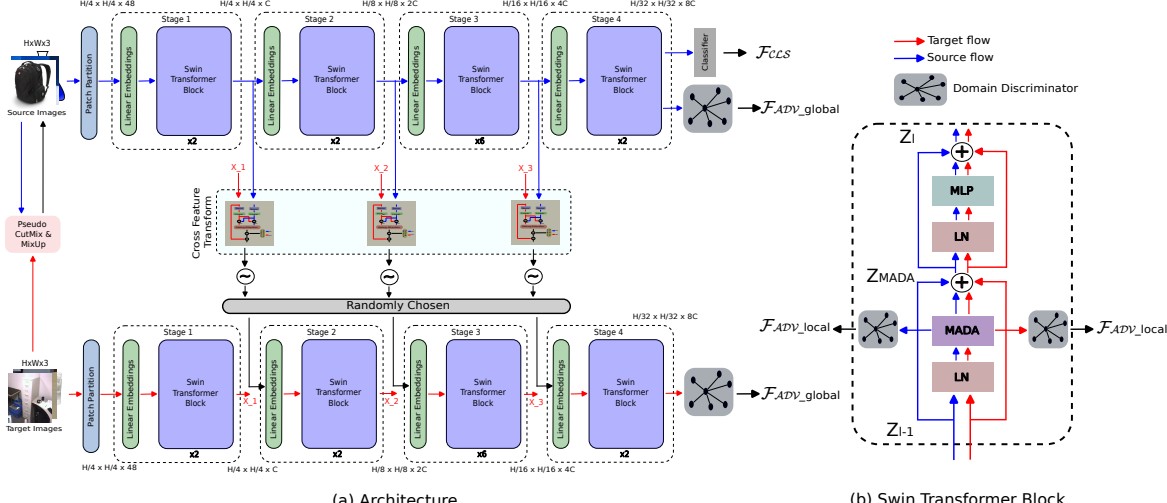

Figure 1: (a) The architecture of the proposed TransAdapter; (b) a Swin Transformer Blocks (notation presented with Eq. 4. MADA is multi-head adaptive double attention module, respectively.

inspired by GANs, have also been widely used in UDA. In these methods, an encoder is trained to generate domain-invariant features by deceiving a domain discriminator, making it unable to distinguish between the source and target domains (Goodfellow et al., 2014; Tzeng et al., 2017). This adversarial process encourages the model to learn features that generalize well across different domains, despite the domain shift.

## 2.2 UDA WITH VISION TRANSFORMERS

While transformers have gained popularity in NLP, their application in UDA for vision tasks is still in its early stages. Some recent work has integrated transformers into CNNs to improve domain adaptation, focusing on critical regions of images (Xu et al., 2021a; Yang et al., 2021b). For example, methods such as cross-attention have been used to blend source and target image representations (Chen et al., 2021a), while others employ multibranch architectures to leverage self-attention and cross-attention mechanisms for feature learning and domain alignment (Saito et al., 2019). The Swin Transformer has also been explored in the context of UDA, where its ability to model local and global relationships in images is harnessed for domain adaptation. However, most of these methods require additional components or specific training strategies to prevent model collapse in challenging tasks (Liu et al., 2022; Yang et al., 2021a).

## 3 METHOD

This section introduces three key modules: the Graph Domain Discriminator (GDD), Adaptive Double Attention (ADA), and Cross-Feature Transform (CFT). GDD models domain relationships using graphs and attention, ADA enhances feature alignment (Deng et al., 2024) via double attention (Zhang et al., 2022), and CFT boosts feature transfer with cross-attention and dynamic gating. Together, they enable efficient domain alignment. The overall architecture is demonstrated in Figure 1.

### 3.1 GRAPH DOMAIN DISCRIMINATOR

The proposed unsupervised domain adaptation method introduces a *Graph Domain Discriminator* (GDD), as shown in Figure 2. The GDD enhances both local and global adaptation by utilizing structural relationships between samples from source and target domains.

For local adaptation, the GDD leverages key features from the third transformer block's attention mechanisms to capture fine-grained details. This includes processing features from windows and

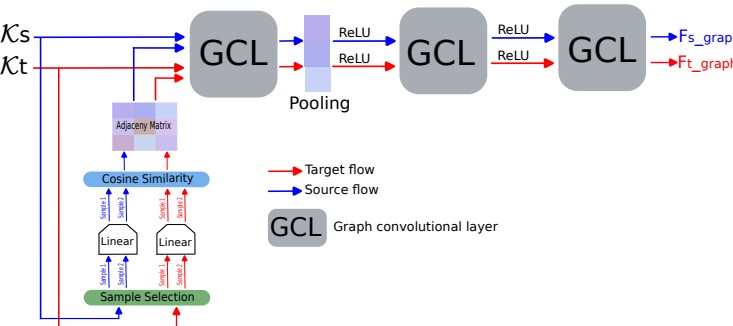

Figure 2: The architecture of the Graph Domain Discriminator uses $K_s$ and $K_t$ to represent source and target key features of MADA, respectively.

shifted windows to align local characteristics, illustrated in Figure 1. For global adaptation, it uses the MLP output from the final transformer block to capture abstract global representations, also depicted in Figure 1. By integrating these strategies, the GDD aligns deep and shallow features, improving the model's generalization across domains.

A vital element of the GDD is the adjacency matrix, which represents the graph structure for convolution operations. It begins with two samples selected by a learnable parameter during training, which are processed through a projection layer $P(\mathbf{x}_{sample_i})$ and $P(\mathbf{x}_{sample_j})$ to fit the adjacency matrix's requirements. The adjacency between samples $i$ and $j$ is defined by cosine similarity:

$$\mathbf{A}_{ij} = \cos(\theta_{ij}) = \frac{P(\mathbf{x}_{sample_i}) \cdot P(\mathbf{x}_{sample_j})}{\|P(\mathbf{x}_{sample_i})\|\|P(\mathbf{x}_{sample_j})\|} \tag{1}$$

Here, $\mathbf{A}_{ij}$ is the adjacency matrix element for samples $i$ and $j$, and $\theta_{ij}$ is the angle between the projected vectors.

The adjacency matrix $A$ is utilized in three layers of graph convolution, each followed by a ReLU activation, facilitating the aggregation of information from individual samples and their neighbors. After the first graph convolutional layer, a pooling operation is applied to reduce the dimensionality and focus on the most salient features, enabling more efficient information processing. To promote domain invariance, a Gradient Reversal Layer (GRL) is introduced, establishing a min-max game between the feature extractor and the domain discriminator. The feature extractor aims to generate features that confuse the discriminator, maximizing the discrepancy for the domain discriminator while minimizing it for the main task, ultimately encouraging the learning of domain-invariant features (Ganin & Lempitsky, 2015).

## 3.2 ADAPTIVE DOUBLE ATTENTION

The adaptive double attention module, shown in Figure 3, processes *windows attention* and *shifted windows attention* simultaneously in each transformer block, termed double attention. It employs a cross-attention mechanism between windows and shifted windows features to enhance alignment and interaction, improving long-range dependency capture and adaptation through dynamic attention re-weighting.

Initially, the module performs feature correction on target domain data to address discrepancies with source features, utilizing a correction block as proposed in (Li et al., 2021a). This block, depicted in Figure 3, modifies the target representation $C^l(F_{x_t, x_{t\_shift}})$ to align it more closely with the source representation, thereby reducing the domain gap and enhancing adaptation.

The correction block consists of two fully connected (FC) layers with ReLU activations. The output of the correction block, $\Delta C^l(F_{x_t, x_{t\_shift}})$, adjusts the representation of the target feature according to the following equation:

$$\hat{C}^l(F_{x_t, x_{t\_shift}}) = C^l(F_{x_t, x_{t\_shift}}) + \Delta C^l(F_{x_t, x_{t\_shift}}) \tag{2}$$

The aim is to make the modified target representation $\hat{C}^l(F_{x_t, x_{t\_shift}})$ similar to the source representation. By doing so, domain alignment is facilitated between $\hat{C}^l(F_{x_t, x_{t\_shift}})$, ensuring that the correction block effectively captures and adjusts discrepancies in the target data to improve the overall adaptation process.

After feature correction, the transformer's attention mechanism processes two feature sets: *windows attention features* and *shifted windows attention features*, represented as $Q, K, V$ and $Q_{shift}, K_{shift}, V_{shift}$, respectively. These matrices correspond to the *query*, *key*, and *value* components of both attention mechanisms (Zhang et al., 2022). By processing them in parallel, the model captures long-range dependencies, enhancing spatial relationship understanding across the image.

Self-attention focuses on windows features, calculating attention via the scaled dot product of $Q$ and $K$, normalized by feature dimension $d$. This allows for local dependency enhancement. Cross-attention combines the query from windows attention ($Q$) with the shifted windows key ($K_{shift}$) to capture complex spatial relationships. Additionally, both attention types are reweighted using entropy-based scaling to prioritize transferable features and suppress domain-specific ones, further improving long-range dependency processing.

$$
\begin{aligned}
A &= \frac{QK^T}{\sqrt{d}} \odot H(F_{graph}) \\
A_{shift} &= \frac{QK_{shift}^T}{\sqrt{d}} \odot H(F_{graph})
\end{aligned}
\tag{3}
$$

Then, the two attention score matrices are concatenated and normalized using the softmax function to produce attention weights (Deng et al., 2024). These final attention weights are applied to the concatenated *value* matrices $[V; V_{shift}]$ to produce the output:

$$
\begin{aligned}
MADA &= \text{Softmax}(\text{Concat}(A, A_{shift})) \times [V; V_{shift}] \\
Z_{MADA} &= MADA(\text{LN}(Z_{l-1})) + Z_{l-1} \\
Z_l &= \text{MLP}(\text{LN}(Z_{MADA})) + Z_{MADA}
\end{aligned}
\tag{4}
$$

Here, $Z_{l-1}$ represents the input to the Transformer block, $Z_{MADA}$ is output of attention, and $Z_l$ is the output after the multi-head adaptive double attention and feed-forward layers. In the context of the Swin Transformer block, layer normalization (LN) is applied to the input of both the attention and MLP layers. Residual connections are added after each operation, ensuring efficient information flow. This architecture allows the model to effectively learn long-range dependencies through shifted window attention.

By integrating both windows and the shifted windows attention mechanisms and applying entropy-based reweighting, the module captures multiple aspects of the feature representations. This process improves long-range dependencies and allows more effective adaptation across domains, improving overall domain adaptation performance, as supported by (Yang et al., 2023).

The module employs an entropy-based reweighting strategy for both self-attention and cross-attention, reweighting the attention scores using the entropy derived from the *graph domain discriminator* output features. The *entropy function* is defined as:

$$
H(F_{graph}) = -\sum_i F_{graph} \log(F_{graph})
\tag{5}
$$

The key and shifted key features of the transformer are first processed by the graph domain discriminator, producing outputs $F_{s_{graph}}$ and $F_{t_{graph}}$. These outputs are then used to calculate entropy, allowing the module to dynamically reweight attention by emphasizing or suppressing features based on domain-specific importance. This enhances adaptation in unsupervised domain adaptation (Yang et al., 2023) and improves the model's generalization across different domains.

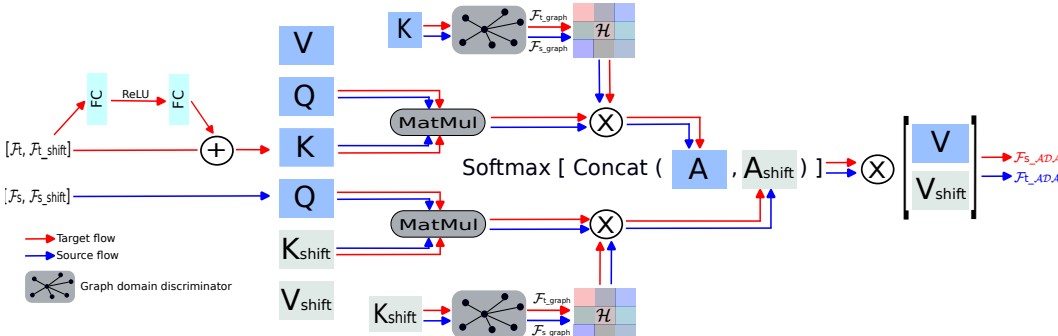

Figure 3: The architecture of Adaptive Double Attention (ADA) module.

## 3.3 CROSS FEATURE TRANSFORM

The proposed Cross Feature Transform (CFT) module enhances domain adaptation within the Transformer architecture by facilitating effective feature alignment between source and target domains. Unlike static methods, the CFT module is applied dynamically after a randomly selected transformer block in each iteration, providing a robust feature transformation approach and reducing the likelihood of overfitting (Sun et al., 2022). The general architecture of the CFT module is illustrated in Figure 4.

Central to the CFT module are bidirectional cross-attention mechanisms, which optimize feature transferability between domains, enabling implicit mixing of features. This enhances the model's ability to learn domain-invariant representations, thereby improving generalization to the target domain (Wang et al., 2022b). The computation of source-to-target attention features $F_{s2t}$ and target-to-source attention features $F_{t2s}$ is performed as follows:

$$
\begin{aligned}
F_{s2t} &= \text{Softmax}\left(f(X_s)^\top g(X_t)\right) \\
F_{t2s} &= \text{Softmax}\left(g(X_t)^\top f(X_s)\right)
\end{aligned}
\tag{6}
$$

To refine feature alignment, the CFT module incorporates a gating mechanism using a learnable parameter $\gamma$, balancing contributions from both directions:

$$
\text{Attn}_{gating} = (1 - \sigma(\gamma)) \cdot F_{s2t} + \sigma(\gamma) \cdot F_{t2s}
\tag{7}
$$

where $\sigma(\gamma)$ is the sigmoid function. This adaptive formulation allows prioritization of source-to-target or target-to-source transformations based on data context.

The pairwise distance between features is computed and combined with the gating attention output:

$$
F_{out} = \left(\text{Attn}_{gating} \times \|F_{s2t} - F_{t2s}\|_2^2\right) + X_t
\tag{8}
$$

Here, $\|F_{s2t} - F_{t2s}\|$ represents the pairwise distance, $\text{Attn}_{gating}$ the gating attention output, and $X_t$ is the target feature added as a shortcut.

## 4 EXPERIMENTS

### 4.1 DATASETS

We utilize four widely recognized benchmark datasets for our experiments: Office31 (Saenko et al., 2010), Office-Home (Venkateswara et al., 2017), and VisDA-2017 (Peng et al., 2017). Following

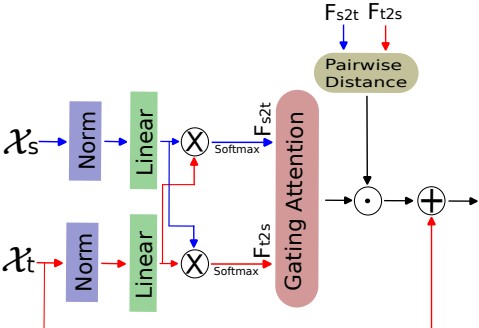

Figure 4: The architecture of Cross Feature Transform (CFT) module. $x_s$ and $x_t$ represents source and target feature, respectively.

the methodology in (Long et al., 2018), we create transfer tasks across these datasets. The Office-31 dataset comprises 4,652 images across 31 categories, divided into three domains: Amazon (A), DSLR (D), and Webcam (W), each sourced from different environments. The Office-Home dataset features images from four domains: Artistic (Ar), Clip Art (Cl), Product (Pr), and Real-World (Rw), each containing 65 categories, thus providing a diverse evaluation benchmark. Lastly, the VisDA-2017 dataset, utilized in the 2018 VisDA challenge, focuses on a synthesis-to-real object recognition task with 12 categories, containing 152,397 synthetic images for the source domain and 55,388 real-world images for the target domain.

### 4.2 DATA AUGMENTATION

We employ CutMix (Yun et al., 2019) and MixUp (Zhang, 2017) as pixel-wise augmentation strategies on raw images to improve feature transferability between domains. Although these methods generally necessitate labeled data, our unsupervised domain adaptation task operates without ground truth labels in the target domain. To tackle this issue, we generate pseudo-labels for the target data using a source-only model trained on the source domain. To reduce noise in these pseudo-labels, we implement a confidence threshold based on the model's accuracy, retaining only predictions that exceed this threshold for the augmentation operations. These augmentations are applied solely to the source data, as our network incorporates a Cross Feature Transform (CFT) module that enhances feature transferability between domains, thus diminishing the necessity for direct augmentation on the target data. The pixel-wise CutMix and MixUp operations, guided by high-confidence pseudo-labels, are illustrated in Figure 1.

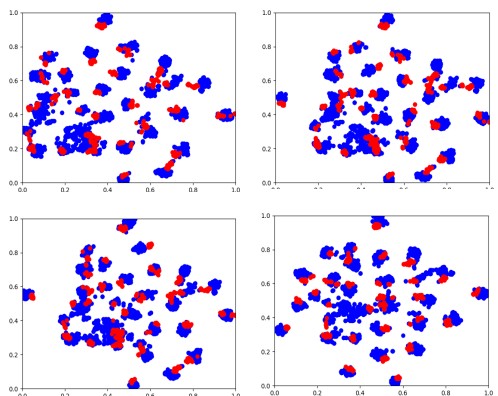

Figure 5: t-SNE visualization of Office-Home dataset, where red and blue points indicate the source and the target domain, respectively. (top left) Swin-B, (top right) +GDD, (bottom left) +CFT, (bottom right) +ADA (TransAdapter)

### 4.3 IMPLEMENTATION DETAILS

For all domain adaptation (DA) tasks, we utilize the Swin-B model, pretrained on the ImageNet dataset (Deng et al., 2009), as the backbone network in our proposed TransAdapter method, integrating 12 dual transformer blocks from Swin-B within the TransAdapter framework. The model is optimized using the Stochastic Gradient Descent (SGD) algorithm (Bottou, 2010), with a momentum of 0.9 and a weight decay parameter of $1 \times 10^{-3}$. We employ a base learning rate of $1 \times 10^{-2}$ for the Office-31, and Office-Home datasets, while a lower learning rate of $1 \times 10^{-3}$ is

applied for the VisDA-2017 dataset. The learning rate follows a warmup cosine scheduler, gradually increasing during the initial training phase and subsequently decaying throughout the remaining iterations. Across all datasets, the batch size is consistently set to 32, and the model is trained over $15,000$ iterations. The hyperparameters $\lambda_{\text{local}}$, and $\lambda_{\text{global}}$ in the TransAdapter method are set to $0.1$, and $0.01$, respectively, for all DA tasks, as shown in Equation 11.

## 4.4 OBJECTIVE FUNCTION

Our domain adaptive model's objective function combines cross-entropy loss for classification, local adaptation loss (strong alignment), and global adaptation loss (weak alignment) The classification layer is a single fully connected layer. For the labeled source domain, the cross-entropy loss is defined as:

$$L_{\text{cls}} = \text{CE}(G(F_{cls}), y_s) \tag{9}$$

where $G(\cdot)$ normalizes and flattens the transformer features, $y_s$ is the ground truth for the source data, and $\text{CE}(\cdot, \cdot)$ denotes the cross-entropy loss, using only the source domain features $F_{cls}$.

**Local-Global Adaptation Loss**: The combined loss function is computed by averaging the cross-entropy loss for local adaptation and the focal loss for global adaptation across both source and target domains:

$$
\begin{aligned}
L_{local} &= \frac{1}{2}\left(\text{CE}(G(F_{ADV\_local}^{src}), \hat{y}^{src}) + \text{CE}(G(F_{ADV\_local}^{tgt}), \hat{y}^{tgt})\right) \\
L_{global} &= \frac{1}{2}\left(FL(G(F_{ADV\_global}^{src}), \hat{y}^{src}) + FL(G(F_{ADV\_global}^{tgt}), \hat{y}^{tgt})\right)
\end{aligned}
\tag{10}
$$

Where, $\hat{y}^{src}$ and $\hat{y}^{tgt}$ denote the ground truth labels for source and target data, respectively. Specifically, $\hat{y}^{src}$ is set to 1 for source data and $\hat{y}^{tgt}$ is set to 0 for target data. The terms $F_{ADV\_global}$ and $F_{ADV\_local}$ are illustrated in Figure 1. The function $G(\cdot)$ refers to a flattening operation followed by a fully connected layer. CE represents cross-entropy loss and $FL(\cdot)$ represents focal loss to address class imbalance by down-weighting the contribution of easy-to-classify examples.

The overall objective function is:

$$\mathcal{L}_{\text{total}} = \lambda_{\text{local}}\mathcal{L}_{\text{local}} + \lambda_{\text{global}}\mathcal{L}_{\text{global}} + \mathcal{L}_{\text{classifier}} \tag{11}$$

where $\lambda_{\text{local}}$ and $\lambda_{\text{global}}$ are the coefficients for the respective loss components.

## 4.5 RESULTS OF OBJECT RECOGNITION

Table 2, 3 and 1 present the accuracy results on the Office-31 (Saenko et al., 2010), Office-Home (Venkateswara et al., 2017), and VisDA-2017 (Peng et al., 2017) datasets, respectively. Comparisons are made across multiple backbones, including AlexNet, ResNet, DeiT, Swin, and ViT, with methods such as Source Only, DDC (Tzeng et al., 2014b), DAN (Long et al., 2015a), RevGrad (Ganin & Lempitsky, 2015), FFAN (Chen et al., 2019), TAT (Liu et al., 2019), SHOT (Liang et al., 2020), ALDA (Chen et al., 2020), CDTrans (Xu et al., 2021b), BCAT (Wang et al., 2022b), WinTR (Ma et al., 2021), TVT (Yang et al., 2023), and the proposed TransAdapter.

In Table 2 on the Office-31 (Saenko et al., 2010) dataset, the Swin backbone achieves state-of-the-art performance, particularly excelling in the W $\rightarrow$ D task, where it matches the highest accuracy of $100\%$. In particular, the proposed TransAdapter model outperforms BCAT (Wang et al., 2022b) in most tasks, leading to an overall average accuracy of $95.5\%$, compared to BCAT (Wang et al., 2022b) $95.0\%$.

Table 3 presents the results on the Office-Home (Venkateswara et al., 2017) dataset, where TransAdapter again achieves superior performance with the Swin backbone. It achieves the

Table 1: Accuracy (%) on the VisDA-2017 dataset. "-B" indicates that the backbone is Base, respectively. The best performance is marked as bold.

| Method | plane | bcycl | bus | car | house | knife | mcycl | person | plant | sktbrd | train | truck | Avg |
|---|---|---|---|---|---|---|---|---|---|---|---|---|---|
| **ResNet** Backbone | | | | | | | | | | | | | |
| Source Only | 55.1 | 53.3 | 61.9 | 59.1 | 80.6 | 17.9 | 79.7 | 31.2 | 81.0 | 26.5 | 73.5 | 8.5 | 52.4 |
| RevGrad Ganin & Lempitsky (2015) | 81.9 | 77.7 | 82.8 | 44.3 | 81.2 | 29.5 | 65.1 | 28.6 | 51.9 | 54.6 | 82.8 | 7.8 | 57.4 |
| MCD Saito et al. (2018) | 87.0 | 60.9 | 83.7 | 64.0 | 88.9 | 79.6 | 84.7 | 76.9 | 88.6 | 40.3 | 83.0 | 25.8 | 71.9 |
| ALDA Chen et al. (2020) | 93.8 | 74.1 | 82.4 | 69.4 | 90.6 | 87.2 | 89.0 | 67.6 | 93.4 | 76.1 | 87.7 | 22.2 | 77.8 |
| DTA Lee et al. (2019) | 93.7 | 82.2 | 85.6 | 83.8 | 93.0 | 81.0 | 90.7 | 82.1 | 95.1 | 78.1 | 86.4 | 32.1 | 81.5 |
| SHOT Liang et al. (2020) | 94.3 | 88.5 | 80.1 | 57.3 | 93.1 | 94.9 | 80.7 | 80.3 | 91.5 | 89.1 | 86.3 | 58.2 | 82.9 |
| **DeiT** Backbone | | | | | | | | | | | | | |
| Source Only-B | 97.7 | 48.1 | 86.6 | 61.6 | 78.1 | 63.4 | 94.7 | 10.3 | 87.7 | 47.7 | 94.4 | 35.5 | 67.1 |
| CDTrans-B Xu et al. (2021b) | 97.1 | 90.5 | 82.4 | 77.5 | 96.6 | 96.1 | 93.6 | 88.6 | 97.9 | 86.9 | 90.3 | 62.8 | 88.4 |
| **Swin** Backbone | | | | | | | | | | | | | |
| Source Only | 98.7 | 63.0 | 86.7 | 68.5 | 94.6 | 59.4 | 98.0 | 22.0 | 81.9 | 91.4 | 96.7 | 25.7 | 73.9 |
| BCAT Wang et al. (2022b) | **99.1** | 91.6 | 86.6 | 72.3 | 98.7 | **97.9** | 96.5 | 82.3 | 94.2 | 96.0 | **93.9** | 61.3 | 89.2 |
| **TransAdapter (ours)** | 98.6 | **94.1** | **88.3** | **75.2** | **98.9** | 97.2 | **97.9** | **87.1** | **96.8** | **97.7** | 93.2 | **67.6** | **91.2** |
| **ViT** Backbone | | | | | | | | | | | | | |
| Source Only | 98.2 | 73.0 | 82.5 | 62.0 | 97.3 | 63.5 | 96.5 | 29.8 | 68.7 | 86.7 | 96.7 | 23.7 | 73.2 |
| TVT Yang et al. (2023) | 97.1 | 92.9 | 85.3 | 66.4 | 97.1 | 97.1 | 89.3 | 75.5 | 95.0 | 94.7 | 94.5 | 55.1 | 86.7 |

Table 2: Accuracy (%) on the Office-31 dataset. "-S" and "-B" indicates that the backbone is Small and Base, respectively. The best performance is marked as bold.

| Method | $A \rightarrow W$ | $D \rightarrow W$ | $W \rightarrow D$ | $A \rightarrow D$ | $D \rightarrow A$ | $W \rightarrow A$ | Avg |
|---|---|---|---|---|---|---|---|
| **AlexNet** Backbone | | | | | | | |
| Source Only | 61.6 | 95.4 | 99.0 | 63.8 | 51.1 | 49.8 | 70.1 |
| DDC Tzeng et al. (2014b) | 61.8 | 95.0 | 98.5 | 64.4 | 52.1 | 52.2 | 70.6 |
| DAN Long et al. (2015a) | 68.5 | 96.0 | 99.0 | 67.0 | 54.0 | 53.1 | 72.9 |
| RevGrad Ganin & Lempitsky (2015) | 73.0 | 96.4 | 99.2 | 72.3 | 53.4 | 51.2 | 74.3 |
| FFAN Chen et al. (2019) | 83.0 | 99.0 | 99.9 | 76.3 | 63.3 | 60.8 | 80.4 |
| **ResNet** Backbone | | | | | | | |
| Source Only | 68.4 | 96.7 | 99.3 | 68.9 | 62.5 | 60.7 | 76.1 |
| DDC Tzeng et al. (2014b) | 75.6 | 96.0 | 98.2 | 76.5 | 62.2 | 61.5 | 78.3 |
| DAN Long et al. (2015a) | 80.5 | 97.1 | 99.6 | 78.6 | 63.6 | 62.8 | 80.4 |
| RevGrad Ganin & Lempitsky (2015) | 82.0 | 96.9 | 99.1 | 79.7 | 68.2 | 67.4 | 82.2 |
| TAT Liu et al. (2019) | 92.5 | 99.3 | 100.0 | 93.2 | 73.1 | 72.1 | 88.4 |
| SHOT Liang et al. (2020) | 90.1 | 98.4 | 99.9 | 94.0 | 74.7 | 74.3 | 88.6 |
| ALDA Chen et al. (2020) | 95.6 | 97.7 | 100.0 | 94.0 | 72.2 | 72.5 | 88.7 |
| **DeiT** Backbone | | | | | | | |
| Source Only-S | 86.9 | 97.7 | 99.6 | 87.6 | 74.9 | 73.5 | 86.7 |
| CDTrans-S Xu et al. (2021b) | 93.5 | 98.2 | 99.6 | 94.6 | 78.4 | 78.0 | 90.4 |
| Source Only-B | 90.4 | 98.2 | 100.0 | 90.8 | 76.8 | 76.4 | 88.8 |
| CDTrans-B Xu et al. (2021b) | 96.7 | 99.9 | 100.0 | 97.0 | 81.1 | 81.9 | 92.6 |
| **Swin** Backbone | | | | | | | |
| Source Only | 89.2 | 94.1 | 100.0 | 93.1 | 80.9 | 81.3 | 89.8 |
| BCAT Wang et al. (2022b) | **99.2** | **99.5** | 100.0 | 99.6 | 85.7 | 86.1 | 95.0 |
| **TransAdapter** | 99.1 | 98.9 | **100.0** | **99.9** | **88.3** | **87.2** | **95.5** |
| **ViT** Backbone | | | | | | | |
| Source Only | 89.2 | 98.9 | 100.0 | 88.8 | 80.1 | 79.8 | 89.5 |
| TVT Yang et al. (2023) | 96.4 | 99.4 | 100.0 | 96.4 | 84.9 | 86.1 | 93.9 |

highest average accuracy of 88.3%, outperforming BCAT (Wang et al., 2022b) and other methods, particularly in the tasks A → PA and A → AC, where it records 91.8% and 91.5%, respectively.

Finally, Table 1 highlights the performance on the VisDA-2017 (Peng et al., 2017) dataset. Here, the TransAdapter with the Swin backbone significantly outperforms other methods, achieving the highest average accuracy of 91.2%. It shows remarkable performance in challenging categories such as knife, bcycl, and train, where it meets or exceeds the accuracy of existing state-of-the-art methods.

## 4.6 ABLATION STUDY

Table 4 presents the results of the ablation study, showcasing the impact of each module on domain adaptation performance. The baseline *Source Only* model achieved an average accuracy of 81.1%.

Table 3: Accuracy (%) on the Office-Home dataset. "-S" and "-B" indicates that the backbone is Small and Base, respectively. The best performance is marked as bold.

| Method | $A\to C$ | $A\to P$ | $A\to R$ | $C\to A$ | $C\to P$ | $C\to R$ | $P\to A$ | $P\to C$ | $P\to R$ | $R\to A$ | $R\to C$ | $R\to P$ | Avg |
|---|---|---|---|---|---|---|---|---|---|---|---|---|---|
| **AlexNet** Backbone | | | | | | | | | | | | | |
| Source Only | 26.4 | 32.6 | 41.3 | 22.1 | 41.7 | 42.1 | 20.5 | 20.3 | 51.1 | 31.0 | 27.9 | 54.9 | 34.3 |
| DAN Long et al. (2015a) | 31.7 | 43.2 | 55.1 | 33.8 | 48.6 | 50.8 | 30.1 | 35.1 | 57.7 | 44.6 | 39.3 | 63.7 | 44.5 |
| RevGrad Ganin & Lempitsky (2015) | 36.4 | 45.2 | 54.7 | 35.2 | 51.8 | 55.1 | 31.6 | 39.7 | 59.3 | 45.7 | 46.4 | 65.9 | 47.3 |
| **ResNet** Backbone | | | | | | | | | | | | | |
| Source Only | 34.9 | 50.0 | 58.0 | 37.4 | 41.9 | 46.2 | 38.5 | 31.2 | 60.4 | 53.9 | 41.2 | 59.9 | 46.1 |
| DAN Long et al. (2015a) | 43.6 | 57.0 | 67.9 | 45.8 | 56.5 | 60.4 | 44.0 | 43.6 | 67.7 | 63.1 | 51.5 | 74.3 | 56.3 |
| RevGrad Ganin & Lempitsky (2015) | 45.6 | 59.3 | 70.1 | 47.0 | 58.5 | 60.9 | 46.1 | 43.7 | 68.5 | 63.2 | 51.8 | 76.8 | 57.6 |
| SHOT Liang et al. (2020) | 57.1 | 78.1 | 81.5 | 68.0 | 78.2 | 78.1 | 67.4 | 54.9 | 82.2 | 73.3 | 58.8 | 84.3 | 71.8 |
| **DeiT** Backbone | | | | | | | | | | | | | |
| Source Only-S | 55.6 | 73.0 | 79.4 | 70.6 | 72.9 | 76.3 | 67.5 | 51.0 | 81.0 | 74.5 | 53.2 | 82.7 | 69.8 |
| CDTrans-S Xu et al. (2021b) | 60.0 | 79.5 | 82.4 | 75.6 | 81.0 | 82.3 | 72.5 | 56.7 | 84.4 | 77.0 | 59.1 | 85.5 | 74.7 |
| Source Only-B | 61.8 | 79.5 | 84.3 | 75.4 | 78.8 | 81.2 | 72.8 | 55.7 | 84.4 | 78.3 | 59.3 | 86.0 | 74.8 |
| CDTrans-B Xu et al. (2021b) | 68.8 | 85.0 | 86.9 | 81.5 | 87.1 | 87.3 | 79.6 | 63.3 | 88.2 | 82.0 | 66.0 | 90.6 | 80.5 |
| WinTR-S Ma et al. (2021) | 65.3 | 84.1 | 85.0 | 76.8 | 84.5 | 84.4 | 73.4 | 60.0 | 85.7 | 77.2 | 63.1 | 86.8 | 77.2 |
| **Swin** Backbone | | | | | | | | | | | | | |
| Source Only | 64.5 | 84.8 | 87.6 | 82.2 | 84.6 | 86.7 | 78.8 | 60.3 | 88.9 | 82.8 | 65.3 | 89.6 | 79.7 |
| BCAT Wang et al. (2022b) | 75.3 | 90.0 | **92.9** | 88.6 | 90.3 | **92.7** | 87.4 | 73.7 | 92.5 | 86.7 | 75.4 | **93.5** | 86.6 |
| **TransAdapter** | **77.6** | **91.8** | 92.8 | **91.5** | **92.3** | 92.6 | **90.3** | **77.6** | 92.8 | **87.9** | **79.1** | 92.8 | **88.3** |
| **ViT** Backbone | | | | | | | | | | | | | |
| Source Only | 66.2 | 84.3 | 86.6 | 77.9 | 83.3 | 84.3 | 76.0 | 62.7 | 88.7 | 80.1 | 66.2 | 88.7 | 78.7 |
| TVT Yang et al. (2023) | 74.9 | 86.8 | 89.5 | 82.8 | 88.0 | 88.3 | 79.8 | 71.9 | 90.1 | 85.5 | 74.6 | 90.6 | 83.6 |

Adding the Graph Domain Discriminator (GDD) improved accuracy to $84.0\%$, while incorporating the Cross Feature Transform (CFT) module raised it to $87.4\%$. The complete model, including the Adaptive Double Attention (ADA) module, achieved the highest accuracy of $91.0\%$, highlighting the ADA module's role in capturing long-range dependencies. Figure 5 visualizes the domain separation performance of each module using t-SNE on the Office Home dataset, demonstrating improved domain separation with each addition, particularly with the complete model featuring the $+ADA$ module (TransAdapter).

## 5 CONCLUSION

In this paper, we introduce TransAdapter, a novel framework that leverages the Swin Transformer for Unsupervised Domain Adaptation (UDA). Our approach features three specialized modules: a graph domain discriminator, adaptive double attention, and cross-feature transform, which enhance the Swin Transformer's ability to capture both shallow and deep features while improving long-

| Method | Office-31 | Office-Home | VisDA-2017 | Avg |
|---|---|---|---|---|
| Source Only | 89.8 | 79.7 | 73.9 | 81.1 |
| +GDA | 91.7 | 81.6 | 78.8 | 84.0 |
| +CFT | 93.5 | 84.1 | 84.6 | 87.4 |
| +ADA (TransAdapter) | **95.5** | **87.5** | **90.2** | **91.0** |

Table 4: Ablation study of each module (%). The best performance is marked as bold. Last row corresponds the proposed model.

range dependency modeling. Experimental results on standard UDA benchmarks show that TransAdapter significantly outperforms existing methods and demonstrates robustness against domain shifts. However, the combined use of window and shifted window attention may increase computational complexity, and our current implementation lacks task-specific adaptation mechanisms for detection and segmentation. Future work will focus on extending the model for these applications and exploring ways to reduce computational complexity while maintaining long-range dependency modeling.

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
