# OpenReview forum: "TransAdapter: Vision Transformer for Feature-Centric Unsupervised Domain Adaptation"
_ICLR.cc/2025/Conference — ICLR 2025 Conference Withdrawn Submission_

### Official Review · Reviewer_qWMP · 2024-10-18

**Soundness:** 3
**Presentation:** 2
**Contribution:** 2
**Rating:** 5
**Confidence:** 4

**Summary:**

This work introduces TransAdapter, a Unsupervised Domain Adaptation (UDA) framework based on the Swin Transformer. It includes three key modules: a Graph Domain Discriminator for enhanced domain alignment via pixel-wise correlations, an Adaptive Double Attention Module for capturing long-range dependencies, and a Cross-Feature Transform Module for improving generalization by dynamically transforming transformer blocks. These components collectively enhance feature consistency across domains, achieving state-of-the-art performance on challenging UDA benchmarks.

**Strengths:**

1.	The authors proposed to utilize Swin Transformer for UDA tasks.
2.	The proposed TransAdapter is effective on three UDA benchmarks.

**Weaknesses:**

1.	The paper does not provide sufficient motivation for the proposed components:
*	What is the motivation behind introducing the GDD? How does it compare to the conventional domain discriminator, such as the one used in DANN [a]? Please elaborate on the advantages and insights of using GDD in this context.
*	Why using double attention module? What’s the motivation and insight? Can the authors explain how this module addresses specific challenges in domain adaptation? Additionally, providing further empirical or theoretical evidence to demonstrate the advantages of double attention over single attention in domain adaptation tasks would enhance the quality of the paper.
2.	Most of the comparison methods in Tables 1, 2, and 3 are outdated. It is recommended that the authors include more recent approaches proposed in 2023 and 2024.
3.	The experiments on large-scale datasets, such as DomainNet, are missing.
4.	The paper's structure can be improved; the content corresponding to Table 4 should be in Section 4.6 instead of Section 5.
5.	The formatting of Figures 1, 2, 3, and 4 can be improved. The font sizes in these figures are inconsistent, with some being too large and others too small. Additionally, some elements in these figures require further clarification; for instance, in Figure 3, the origins of ‘V’ and ‘V_shift’ need to be explained.

[a] Ganin et al. Unsupervised domain adaptation by backpropagation. In ICML, 2015.

**Questions:**

Please refer to the weakness part.

---

### Official Review · Reviewer_QYrX · 2024-11-02

**Soundness:** 2
**Presentation:** 2
**Contribution:** 2
**Rating:** 3
**Confidence:** 3

**Summary:**

This work presents a new approach to UDA by using the Swin Transformer. It introduces three key modules: a Graph Domain Discriminator for pixel-wise correlation and domain alignment; an Adaptive Double Attention module to enhance long-range dependencies; and a Cross-Feature Transform for selective feature transformation.

**Strengths:**

The proposed method leverages the Swin Transformer architecture and introduces some modules that enhance feature alignment and domain adaptation, leading to performance enhancement on UDA tasks.

**Weaknesses:**

I have some questions about this paper that need further discussion. Please see them below.

If the authors can address my concerns, I am willing to raise my score.

**Questions:**

1. The related work section is outdated, lacking any references from 2022 onward. I believe the authors need to adequately address and enhance this section.

2. Why using the attention from the third transformer block? (L161)

3. The proposed GDD seems to capture relationships between samples from the source and target domains. Why is GCN employed for this purpose? I would appreciate a detailed explanation and a comparison with other common similarity computation methods.

4. In the experiments, the authors introduced data augmentation. Was a baseline method used? If not, this comparison may not be fair.

5. The baseline methods are outdated, such as CDTrans[1], SDAT[2], and MIC[3].

[1] CDTrans: Cross-domain Transformer for Unsupervised Domain Adaptation

[2] A Closer Look at Smoothness in Domain Adversarial Training

[3] MIC: Masked Image Consistency for Context-Enhanced Domain Adaptation

---

### Official Review · Reviewer_EPuk · 2024-11-04

**Soundness:** 2
**Presentation:** 1
**Contribution:** 2
**Rating:** 3
**Confidence:** 5

**Summary:**

This paper aims to solve the task of unsupervised domain adaptation (UDA) and proposes a method called TransAdapter. This framework consists of three main modules. Graph Domain Discriminator captures pixel-wise correlations through a graph convolutional layer. Adaptive Double Attention processes windows and shifted windows attention to increase long-range dependency features. Cross-Feature Transform transfers source features to target features. The authors conduct experiments on three datasets commonly used for UDA tasks: Office-31, Office-Home, and VISDA-2017. The ablation study in Table 4 provides preliminary verification of the effectiveness of three modules.

**Strengths:**

This paper focuses on UDA and proposes a novel framework, TransAdapter, applied to the Swin Transformer. The paper dedicates a significant portion to detailing the specific structure of the three modules. Experimental results demonstrate that TransAdapter can improve the performance of pre-trained Swin Transformer (Source Only) on UDA tasks.

**Weaknesses:**

1. Overall, the three modules in TransAdapter are specifically designed for Swin Transformer, and they appear difficult to generalize to other models, even those based on Transformer like ViT. Additionally, the paper does not provide any experiments or discussions that can be referenced.

2. The motivation for using Swin Transformer is stated in lines 58-62 of the main text. The relevance of long-range dependencies and non-overlapping patches processing to demonstrate that Swin Transformer is suitable for UDA tasks does not seem strong. The authors need to provide more insights into this motivation.

3. The paper states that the "Graph Domain Discriminator (GDD) leverages key features from the third transformer block’s attention mechanisms to capture fine-grained details," as mentioned in line 161. However, there is no discussion in the paper on why the third block is the best. The authors need to provide additional ablation studies to explain this choice.

4. The authors mention the use of Gradient Reversal Layer (GRL) in lines 194-199. However, they do not explain how GRL is applied in GDD, and there is no explanation elsewhere in the paper, not even in the figures.

5. The references to $f$ and $g$ in Equation 6 are unclear. Section 4.4 (Objective function) is inappropriately placed in the experiment section.

6. In Section 4.2, the authors state that CutMix and MixUp are used as data augmentation strategies. However, there is no ablation study on their effects, making it difficult to discern whether the performance improvement of TransAdapter comes from data augmentation.

7. The paper only uses Swin-B for experiments. At least Swin-T, Swin-S, and Swin-L should be used to demonstrate the reusability of TransAdapter across different Swin Transformer models.

8. The paper lacks ablation studies on hyperparameters, such as $\lambda_{local}$ and $\lambda_{global}$.

**Questions:**

See Weaknesses

---

### Official Review · Reviewer_NmVh · 2024-11-04

**Soundness:** 2
**Presentation:** 2
**Contribution:** 1
**Rating:** 3
**Confidence:** 5

**Summary:**

The paper addresses Unsupervised Domain Adaptation (UDA) in the context of Vision Transformers. Specifically, while several prior works on UDA have leveraged CNNs for adaptation, this work attempts to utilize Vision Transformers (ViTs), especially the Swin transformer architecture for UDA. The authors propose TransAdapter - a transformer-based approach to UDA that draws heavily from the Swin transformer. TransAdapter comprises three components - (i) a domain adversarial Graph Domain Discriminator (GDD) (ii) Adaptive Double Attention that simultaneously processes window and shifted window attentions (iii) Cross-Feature Transform (CFT) to transfer the source to target features. Experiments on the standard UDA benchmarks demonstrate the improvements that TransAdapter achieves over prior works.

**Strengths:**

The paper proposes a novel improvement to the Swin transformer specifically for the UDA task. Experiments on the standard Domain Adaptation benchmarks demonstrates the effectiveness of the proposed method compared to prior works.

**Weaknesses:**

### (a) Presentation and discussions
- **Unclear motivation -** The fundamental motivation behind proposing improvements to the Swin transformer architecture is not concrete. The authors present the argument of using transformers rather than CNNs for UDA based on the improvements that transformers achieve on several vision tasks. However, they do not discuss the limitations of the Swin transformer and how their method improves on the same.

- **Justification of design choices -** While the authors have chosen the Swin transformer architecture as the base and provided an intuitive justification of the same, there is no discussion on the design choices behind various aspects of the method. For example, why does the Graph Domain Discriminator (GDD) use graph convolutions rather than regular convolutions? Why use a domain discriminator to align the source and target representations? Why specifically use the third transformer block’s attention output for the GDD? The authors should discuss all these aspects in detail for a good understanding of TransAdapter.

- **Discussions on each component of TransAdapter -** Even though the authors have provided high-level intuitions behind each component of TransAdapter, this does not convey a clear understanding of the same.
    - For example, in L254-256, the authors mention that integrating the window and shifted window attention mechanism along with reweighting enables the model to capture “multiple aspects of the feature representations”. What aspects does the model capture? How do they help with the UDA task? How are they an improvement over the representations captured by the original Swin transformer architecture?
    - In L223-225, the authors claim that processing the window and shifted window attentions in parallel enables the model to capture long-range dependencies and enhances spatial relationship understanding. There is, however, no discussion on how the Adaptive Double Attention Module achieves any of these aspects. How exactly does the modified attention module capture global dependencies better than a vanilla ViT and a Swin transformer? Moreover, what is long-range dependence in the context of a vision transformer?
    - Overall, the authors should provide detailed discussions on each aspect of the proposed method, and how they improve over the corresponding components in the Swin transformer. The lack of these discussions makes it quite challenging to understand the motivation behind the work.

- **Use of terminologies -** The authors make use of several terminologies without providing a concrete definition or intuition behind the same. For example, the authors specify “global and local adaptation” without explaining these terms. Do they refer to the alignment of global and local features across the source and target domains? What do global and local features refer to in the context of UDA and the Swin transformer? The authors should clearly specify what these terms mean and how they relate to the context of UDA.

### (b) Experiments and results
- **Missing comparisons-** While the authors have compared with several prior works that utilize the ViT backbone, the comparisons are not extensive and complete. Ideally, the authors should extend the ResNet-based methods to Swin or ViT for a fair comparison. The goal is to ensure a fair comparison by comparing the proposed method with prior works using the same / similar architecture. The list of comparisons missing is given below:
    - SHOT and [R2, R3, R5] with ViT
    - SHOT and [R2, R3, R5] with Swin transformer
    - [R1] with DeiT
    - SSRT [R4] with ViT, DeiT and Swin transformer
- **Missing results-** The authors have not presented results on DomainNet, which is the most important and challenging UDA benchmark. They should present results on DomainNet, either in the single source or multi-source / multi-target setting (in case of time and compute constraints) to demonstrate the effectiveness of the approach. Additionally, how does the proposed method perform in a multi-source or multi-target setting?
- **t-SNE visualidations-** There is no discernible difference among the four plots in Fig. 5. Moreover, there are no details provided on the setup for the visualisation experiment. Which layers’ features were considered for visualisation? What is the expected pattern in the plots? What is observed in the plots and what does that indicate about the proposed method? The authors should answer all these questions for a clear analysis of the proposed method with respect to the visualisation.

### (c) Missing references
- R1 - Sanyal, Sunandini, et al. "Domain-specificity inducing transformers for source-free domain adaptation."ICCV 2023.
- R2 - Hoyer, Lukas, et al. "MIC: Masked image consistency for context-enhanced domain adaptation." CVPR 2023
- R3 - Yang, Shiqi, Shangling Jui, and Joost van de Weijer. "Attracting and dispersing: A simple approach for source-free domain adaptation." NeurIPS 2022.
- R4 - Sun, Tao, et al. "Safe self-refinement for transformer-based domain adaptation." CVPR 2022.
- R5 - Zhang, Yixin, Zilei Wang, and Weinan He. "Class relationship embedded learning for source-free unsupervised domain adaptation." CVPR 2023.

**Questions:**

In addition to all the questions mentioned in the weaknesses section, the reviewer has a few follow-up questions.
- What is the motivation behind TransAdapter? Is it intended to be an improvement over the Swin transformer specifically tailored to UDA? If so, what limitations of Swin does TransAdapter overcome?
- What is the purpose of using graph convolutions instead of regular convolutions in the domain discriminator? What is the motivation behind the same?
- What is the exact benefit of simultaneously processing the window and shifted window attentions rather than sequentially in the original Swin transformer? Can the authors demonstrate this through a simple motivating experiment or visualisation?
- How does TransAdapter compare to the original Swin transformer in terms of parameter count, training and inference time?
- Can the proposed method be extended to the multi-source and multi-target settings? How would the proposed method perform in these settings?

---

### Note · Authors · 2024-11-17

**Comment:**

I appreciate for all reviewers' feedback. I am planning to develop my article in terms of all feedbacks

**Withdrawal Confirmation:**

I have read and agree with the venue's withdrawal policy on behalf of myself and my co-authors.